# Potential implications of rising sea level on American Horseshoe Crab (*Limulus polyphemus*) spawning beaches in two Florida counties

Danielle Contrada⊕, Claire Crowley-McIntyre⊕, Berlynna Heres⊕⊕*

Florida Fish and Wildlife Conservation Commission, St. Petersburg, Florida, United States of America

⊕ All authors contributed equally to this work.
* Berlynnah@gmail.com

## Abstract

Coastlines support a diversity of wildlife and are used as spawning sites for many species. Though, as sea levels rise, many coastlines will be inundated, affecting species that live or nest in these habitats. The American Horseshoe Crab (*Limulus polyphemus*) requires specific environmental conditions for optimal egg development in coastal habitats. This study used reports of horseshoe crab sightings in two Florida counties to identify characteristics associated with their spawning habitats. A 0.6-meter-wide shoreline polygon was created in ArcGIS Pro and each habitat type within the polygon was identified. Using sea-level rise scenarios for 2040, 2070, and 2100, the percentage of habitat loss within these spawning polygons was predicted. In Pinellas County, spawning was predominantly associated with sheltered tidal flats and mangroves. In Brevard County, spawning was most often associated with fine- to medium-grain sand beaches and mangroves. These results demonstrated that each sea-level rise scenario projects a change in habitat area within the shoreline polygons, most notably in the 2040 projection. By 2100 in Pinellas County, 96.3% of mangroves and 87.4% of sheltered tidal flats within the shoreline polygons were predicted to be lost or changed to a different habitat type. In Brevard County by 2100, 98% of fine- to medium-grain sand beaches and 94.8% of mangroves within the shoreline polygons were predicted to be lost or changed to a different habitat type.

## Introduction

Over the last century, climate change has caused the global mean sea level to rise at an unprecedented rate. Along the contiguous United States, relative sea level rose approximately 28 centimeters (cm) from 1920–2020, with a notable acceleration after 1970 [1]. Along the Florida coast, sea level is rising 2.5 cm every 11–14 years [2]. Rising sea level poses a threat to coastal communities and can drastically change existing coastal ecosystems. These direct threats include, damage to infrastructure,

**Data availability statement:** The raw survey data is available as supplemental material. The remaining data is open source and the origins are described in the methods section of the manuscript.

**Funding:** The author(s) received no specific funding for this work.

**Competing interests:** The authors have declared that no competing interests exist.

permanent loss of land and life, ecological regime shifts, and decreased water quality [1]. Indirectly, these communities face worsening of impacts of storm surge, high tides, and wave action, both chronic and acute in nature [1]. To combat this threat, various protective measures, such as sea walls, surge barriers, and nature-based defenses such as wetlands, have been implemented [1]. The installation of such structures can negatively impact species, such as the American Horseshoe Crab (*Limulus polyphemus*), hereafter horseshoe crab, as their spawning habitats along the shoreline are compressed between the rising sea and infrastructure [3–5]. Bulkheads and seawalls can obstruct access to intertidal spawning beaches, exacerbate shoreline erosion, and prevent natural beach migration [6]. Riprap, a type of rock armor, can also trap and strand species, leading to increased mortality [7]. Additionally, building anthropogenic structures to protect coastal communities can interfere with natural geological processes that are crucial for preserving habitats such as spawning beaches [8]. The amalgamation of these factors can lead to negative consequences for species such as horseshoe crabs that require specific environmental conditions for reproductive success [2].

The horseshoe crab resides along the Atlantic and Gulf of Mexico coastlines from Louisiana to Maine [2,9], and along the Yucatan Peninsula of Mexico [2]. Although horseshoe crabs are known to tolerate a wide variety of environmental conditions, the rate at which water quality and habitat availability changes occur may weaken their long-term viability [10]. The conservation status of the horseshoe crab varies geographically. Some populations are at risk due to limited and fragmented habitats (Gulf Coast of Maine), while others are considered stable or rising (Mid-Atlantic and Southeast) [2]. There is evidence of mixed trends along Florida's coastline, but the causes of localized population decline in Florida are poorly understood [2]. Across regions, the availability of spawning locations that offer optimal conditions for egg development is a critical factor that can affect the conservation status and long-term viability of horseshoe crab populations.

Horseshoe crabs mature at nine or ten years old and typically begin spawning along sandy shorelines when the tides are highest during the spring and fall [2]. The horseshoe crabs gather in sometimes large congregations, the female crabs dig a small hole in the high tide line of the shore to lay her eggs while an attached male fertilizes them externally [2]. These eggs develop over the course of one month to six weeks, in which time the habitat must remain consistently inundated with water but not fully covered, to prevent the eggs from drying out or becoming anoxic due to extended inundation [10,11]. Although horseshoe crabs will spawn in suboptimal areas if suitable locations are unavailable, they typically choose spawning sites that provide optimal conditions for embryonic development [11]. Generally, the properties of shoreline sediments influence horseshoe crab egg viability and development by affecting moisture retention and the rate of water movement through the beach [12]. Studies have revealed that the grain size of Delaware beaches, which have a high concentration of spawning horseshoe crabs, varies between 0.5 to 2.0 millimeters (mm) in diameter, with a median grain size of 0.7 mm [13]. Beaches in South Carolina and Florida with spawning horseshoe crabs have reported mean grain sizes of

0.2–0.4 mm [14] and 0.3 mm [12], respectively. Wave action is also a factor in optimal development because beaches with high-energy wave action may exhume buried eggs, exposing them to suboptimal moisture and oxygen gradients [15]. Preferred areas are characterized by lower wave energy, which reduces the risk of exposing or washing away horseshoe crab eggs. Just above the mean high-tide line is considered the most suitable place for spawning since sediments at higher elevations are susceptible to desiccation [12]. In comparison, sediments in lower elevations have lower oxygen levels that can impair egg development [12].

The slope of the beach is a determining factor that affects the conditions for the development of eggs. Horseshoe crabs exhibit a preference to spawn on beaches that are sloped 2°–8° to ensure ideal conditions for embryonic development [12,16]. In New Jersey, spawning beaches have recorded slopes in the range of 3°– 7° seaward [17], while in Delaware and along the Gulf Coast of Florida, beach slopes ranged from 2°– 5° [12,16]. Generally, steeper-sloped beaches have larger grain sediment, such as pebbles and boulders, which are unsuitable for egg development [17]. Alternatively, sand and gravel substrates with adequate amounts of aeration and moisture are considered ideal for spawning success [12,15].

The timing of horseshoe crab spawning varies depending on the location. However, it is typically triggered by warmer water temperatures in the spring, which stimulate adult horseshoe crabs to migrate from deep to shallow waters [18]. Florida Horseshoe crabs are unique because they have two distinct spawning periods: February to May and August to November. This is unlike the northern populations from Delaware to Maine, which have one short spawning period, typically in May and June [19].

The timing of spawning also plays an important ecological role, especially in the northern United States, where studies have indicated a relationship between horseshoe crab spawning events and the Red Knot (*Calidris canutus rufa*), a migratory shorebird. The Red Knot relies on horseshoe crab eggs in Delaware Bay to fuel their migration from tropical wintering grounds to Arctic Canada [20]. Several species of crustaceans and fish have been directly observed feeding on horseshoe crab eggs and larvae, and found in the stomach contents of fish in the tidal creeks of New Jersey [2]. Furthermore, while horseshoe crabs feed, their plowing action helps aerate the substrate, which promotes species diversity, richness, and abundance [21].

In addition to their ecological roles, horseshoe crabs have been economically important for many years. From the mid-1800s until the 1960s, they were harvested in Delaware Bay and used primarily for fertilizer [22]. Today, horseshoe crabs are commercially used as bait in the American eel (*Anguilla rostrata*) and whelk (*Busycotypus canaliculatus*) fisheries [23,24], educational and aquarium purposes, and biomedical research [25]. In 2021, the coastwide bait landings included 741,684 horseshoe crabs, which were well below the coastwide quota of 1.59 million. Florida has a unique "marine life" fishery where horseshoe crabs are harvested for aquarium trade and educational and scientific purposes. In the biomedical industry, over 600,000 horseshoe crabs are collected each year [26], and according to the Atlantic States Marine Fisheries Commission [23], 718,809 horseshoe crabs were collected in 2021 and their blood used by the medical industry for contaminant testing. Assessing the threat of sea level rise on critical spawning habitat for horseshoe crabs is essential because of these sectors' economic dependence on robust horseshoe crab populations and the need for mitigation that considers horseshoe crab spawning beach requirements in the design.

This study describes the potential impact of sea-level rise on horseshoe crab spawning habitats in two regions of Florida with the highest recorded sightings. Data from public reports were used to assess what habitats are used by spawning horseshoe crabs in Pinellas County on the Gulf Coast and Brevard County on the Atlantic Coast of Florida. Then, sea level rise scenarios for 2040, 2070, and 2100 were examined to identify and quantify the potential loss or change of spawning habitats due to rising sea levels. While these results are specific to Florida, the methods can be applied to other areas throughout the horseshoe crab's range, specifically in areas where similar public, citizen science, or fishery-independent surveys are routine.

## Methods

### Study sites

To assess the potential change of horseshoe crab (*L. polyphemus*) spawning habitat due to sea level rise in Florida, two highly developed coastal counties, Pinellas and Brevard, were selected for analyses based on their high concentrations of public reported horseshoe crab activity [27]. Pinellas County is located along the Gulf Coast of Florida, with a coastline that spans 946 kilometers. Brevard County is located along the Atlantic coast of Florida, with an estimated 785 kilometers of coastline, including the Indian River Lagoon. Both counties are home to a wide array of habitats, ranging from sandy beaches to mangrove forests. While these sites are geographically separated and the horseshoe crabs are part of genetically distinct populations [25], spawning behaviors were assumed to be similar. The sites were not selected to represent different populations or climate change regimes.

### Horseshoe crab spawning data

Horseshoe crab spawning in Florida has been reported by the public through phone, email, and online surveys, including Survey 123 [28] since 2002 [27]. Beginning in 2017, observations of horseshoe crabs reported through online surveys and Survey 123 (S1 Fig) included GPS locations; therefore, in the present study, observations collected through these methods from 2017–2023 were chosen based on their location accuracy. Horseshoe crab sighting reports from January 2017–June 2023 from Pinellas County and sightings from January 2020–June 2023 from Brevard County were included in the analyses due to the high reporting frequency in these counties. Heat maps were created using a Kernel Density Analysis, which calculated the density of sightings in both counties [29]. It is important to note that subtidal spawning was not considered in this study.

Horseshoe crab sighting reports were examined to distinguish which observations reflected a location where horseshoe crabs can physically nest. All data were reviewed by examining coordinate locations using Google Earth version 10.42.0.2 or in-person accounts. Based on previously studied behavior, a ranking system was established to evaluate the possibility of horseshoe crabs being able to spawn at each reported location and eliminating those areas with a hardened or armored shoreline, which has been shown to prevent active spawning [12,21] (Table 1). Habitats ranked "zero", which classified areas where spawning was possible, were used for further analyses.

### Spawning habitat characterization

ArcGIS Pro version 3.1 (Esri, Redlands, CA, USA) was used to spatially identify critical horseshoe crab spawning locations and their vulnerability to rising sea levels. Data were projected in the coordinate system for Florida using NAD 1983 (2011) and Florida GDL Albers (meters) [30]. The starting dates were selected to coincide with each county's newest Environmental Sensitivity Index (ESI) publications (2016, 2020) [31]. The ESI is a spatial dataset that characterizes the marine and coastal environments according to their sensitivity to oil spills. ESI layers, which are collections of geographic data, were chosen to identify coastal habitat designations in Florida that correlate with horseshoe crab spawning sites. ESI Shorelines are classified by map developers through a combination of overflights, aerial photography, remotely sensed

**Table 1. Description of ranking system used to evaluate each publicly reported horseshoe crab spawning location.**

| Rank | Description |
|------|-------------|
| 0 | Spawning physically possible in this location |
| 1 | Location not possible for spawning |
| 2 | Cannot make sufficient conclusion about spawning site |

data, ground truthing, and existing maps and data spanning the previous 10 years [31]. Although habitats such as riprap and man-made structures are not considered typical horseshoe crab spawning sites, they have been included in this analysis. Where the ESI designated the shoreline of a spawning location as riprap, it was in proximity or adjacent to sandy beach patches where spawning is likely to occur. These sandy patches are smaller than the larger areas of riprap or man-made structures and were therefore classified based on the prominent habitat type in that location. By including habitats such as riprap that are adjacent to areas where horseshoe crabs can spawn, it can be assumed that the decline of such habitats because of rising sea levels may also decrease sand accumulation in these small sandy beach patches.

The ESI was used to characterize the spawning habitat types of horseshoe crabs and to examine potential habitat changes due to sea level rise in Pinellas County and Brevard County, Florida. Although the ESI was created for oil spill response, it is also a robust characterization of coastal habitats that provides a baseline for calculating horseshoe crab spawning habitats. The ESI includes sensitivity rankings, landward and seaward shore type designations, and a generalized ESI type [31]. Shoreline types, particularly seaward designations, were used for these analyses. The newest ESI versions for Florida's "Northwest Peninsula" (2016) and "East Coast" (2020) were projected in ArcGIS Pro and clipped to the Pinellas County and Brevard County extents [31]. The outer edges of the ESI, which include the Atlantic and Gulf Coast facing beaches were removed from coastline measurements as spawning does not occur in these areas due to wave action.

To further elucidate the characteristics of spawning habitats used by horseshoe crabs in Florida, as compared to other areas within the horseshoe crabs range, the Digital Elevation Model for each county was used to determine the slope of these habitats. Slope calculations were established using the "Slope" tool in the spatial analyst toolbox. Subsequently, the "Add Surface Information" tool was used to calculate the median slope of the locations where spawning horseshoe crabs were present. Man-made structures and hardened structures were removed from slope analysis because the slopes reflected a vertical structure and skewed slope medians to exceed 90°.

### Sea-level rise scenarios

In the present study, sea-level rise scenarios were based on the most recent "Intermediate-Low" scenario by the National Oceanic and Atmospheric Administration (NOAA) predictions [1] to determine a conservative estimate of habitat changes in areas most likely used by spawning horseshoe crabs. This scenario predicts the global mean sea level to rise by 0.5 meters by 2100 and includes estimates of vertical land motion and probabilities for extreme water levels along the U.S. coastline.

### Horseshoe crab spawning locations

To establish a polygon of each shoreline type (henceforth, shoreline polygon) and identify a conservative estimate of the shoreline polygon that could be impacted by a 0.5 m sea-level rise, a 0.6 m buffer toward land was created to represent the area of optimal spawning habitat surrounding the high tide line. The ranked horseshoe crab reports were projected in ArcGIS Pro (Esri, Redlands, CA, USA), and a spatial join was created between the horseshoe crab and ESI layers to characterize the seaward shoreline type closest to each sighting. If the seaward designation was uncharacterized, the landward shoreline type was substituted.

### Digital elevation model data

The Digital Elevation Model (DEM) [32] was used in conjunction with the intermediate-low risk scenario [33] to create sea level rise predictions. The DEM is lidar based and the resolution is approximately 3 meters. To visualize these scenarios, the "Raster Calculator" tool was used to create an expression for each timestep (2040, 2070, 2100). The timesteps identified cells in the raster considered "underwater" in a specified year. The thirty year interval was selected to observe an immediate or gradual change using established projections [1]. Next, the "Reclassify" tool was used to remove cells that

were above sea level based on their elevation. Finally, the "Raster to Polygon" tool converted the raster layers for each timestep into polygons for further analyses.

The "Erase" tool was used in conjunction with the shoreline polygon and sea level rise polygon layers to determine the percentage of habitat within the shoreline polygon that would be lost or transition to a different habitat type. The percentage was calculated for each habitat polygon in years 2040, 2070, and 2100.

## Results

### Horseshoe crab spawning data and habitat characterization

**Pinellas County.** Horseshoe crab sightings in Pinellas County varied by location and year. From 2017–2023, there were a total of 250 horseshoe crab spawning sightings (Fig 1). In 2017, there were 17 Horseshoe crab spawning reports, followed by 34 in 2018, 56 in 2019, 53 in 2020, 15 in 2021, 27 in 2022, and 48 in 2023, which included reports through June (Fig 1).

Based on a Kernel Density analysis, there were three main clusters of points around Honeymoon Island, St. Petersburg, and Fort De Soto Park (Fig 2). Peak Horseshoe crab sightings were found during March (n = 45) and April (n = 46).

The habitats associated with spawning horseshoe crabs in Pinellas County varied, with the highest proportion in sheltered tidal flats (41.2), followed by mangroves (24), exposed tidal flats (16.8), course-grain sand beaches (6.4), fine- to medium-grain sand beaches (5.6), and mixed sand and gravel beaches (2.4) (Fig 3). When considering the landward habitat association, 23% of sightings were adjacent to man-made structures. In Pinellas County, the median slope of all spawning habitats calculated together was 0.9° (minimum = 0°; maximum = 19.8°). When isolating the beach habitats (fine to medium-grain sand, course-grained sand, and mixed sand and gravel beaches), the median slope is 2° (minimum = 0°; maximum = 9.7°).

### Brevard County

Brevard County had 278 horseshoe crab spawning sightings from 2020–2023, which varied by location and year (Fig 1). In 2020, there were 112 horseshoe crab spawning reports, followed by 27 in 2021, 77 in 2022, and 62 in 2023, which includes reports through June 2023 (Fig 1). In general, spawning reports peaked in February (n = 83) and March (n = 77). Based on a Kernel Density analysis, there were two main clusters of points near Titusville, and Port Canaveral (Fig 4).

There were six habitats with the highest proportion of spawning horseshoe crab sightings in Brevard County (Fig 5): fine to medium-grain sand beaches (29.1), mangroves (17.6), scrub-shrub wetlands (16.6), vegetated low banks (11.5), riprap (7.6), and sheltered riprap (5). When considering the landward habitat association, 17% of sightings were adjacent to man-made structures. In Brevard County, the median slope of all spawning habitats calculated together was 0° (minimum = 0°; maximum = 19.1°). When isolating the beach habitats (fine- to medium-grain sand, course-grain sand, and mixed sand and gravel beaches), the median slope is 2.7° (minimum = 0°; maximum = 19.1°).

### Sea–level rise scenarios

Sea level rise projections for 2040 showed a significant loss or change in habitat types for the shoreline polygons created in both Pinellas and Brevard Counties. The total area of Pinellas County shoreline polygons for 2016 was 819,763.16 m², and the total area of Brevard County shoreline polygons for 2020 was 838,797.60 m², based on the assumed 0.6 m buffer of the landward designation (Fig 6 and Fig 7). Mangrove habitats in Pinellas County had the most significant percent loss within the polygon area by 2040 (90%). For Brevard County, all beach habitats including mixed sand and gravel, course-grained sand, and fine-medium grained sand had the most significant percent loss by 2040 (95%, 92%, 91.9%, respectively). Both counties show a slight decrease in the rate of habitat loss between 2040 and 2100.

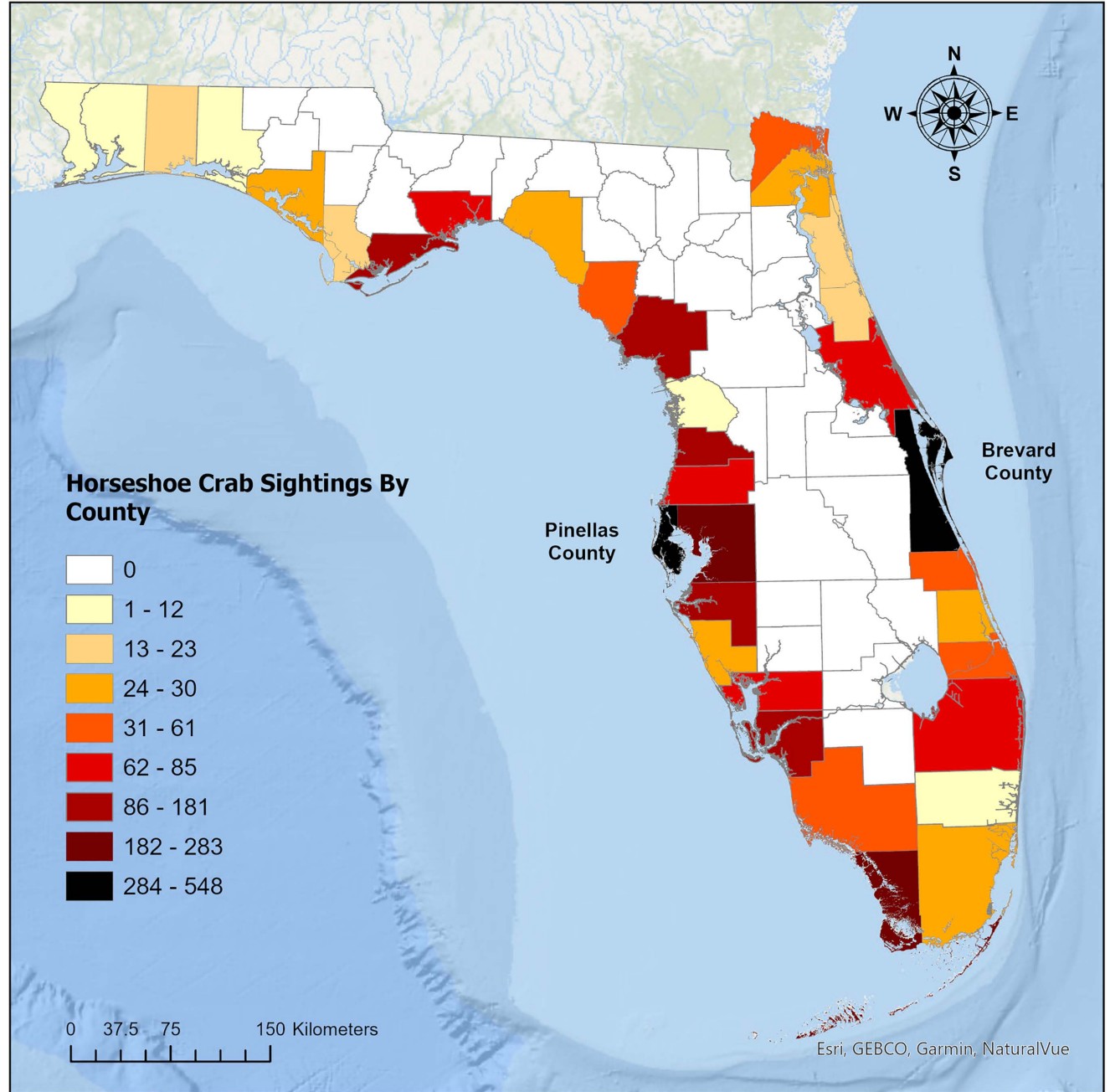

**Fig 1. Cumulative public reports of spawning horseshoe crabs by county in Florida from January 2002– June 2023 (N = 6,752).** Darker colors represent a higher concentration of reports.

The largest loss in Pinellas County shoreline habitats associated with horseshoe crab spawning occurs from 2016 to 2100 (Table 2). Within the shoreline polygons, the habitat losing the most area was mangroves (96.3%), followed by sheltered tidal flats (87.4%), exposed tidal flats (75%), course-grain sand beaches (63.2%), and fine- to medium-grain sand beaches (56.6%).

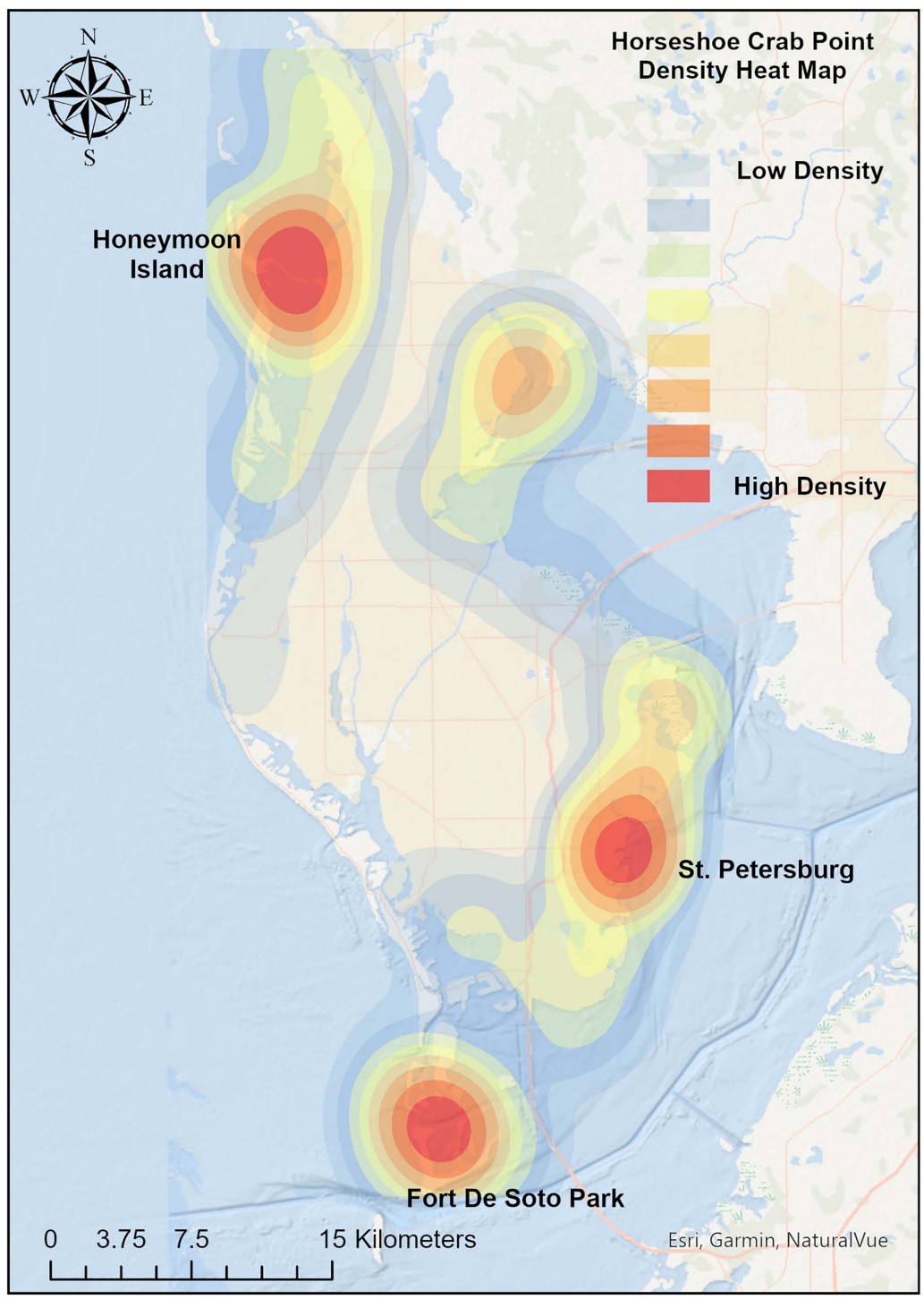

**Fig 2. Horseshoe crab spawning locations in Pinellas County, Florida.** Color denotes density of sightings in each location from 2017–2023.

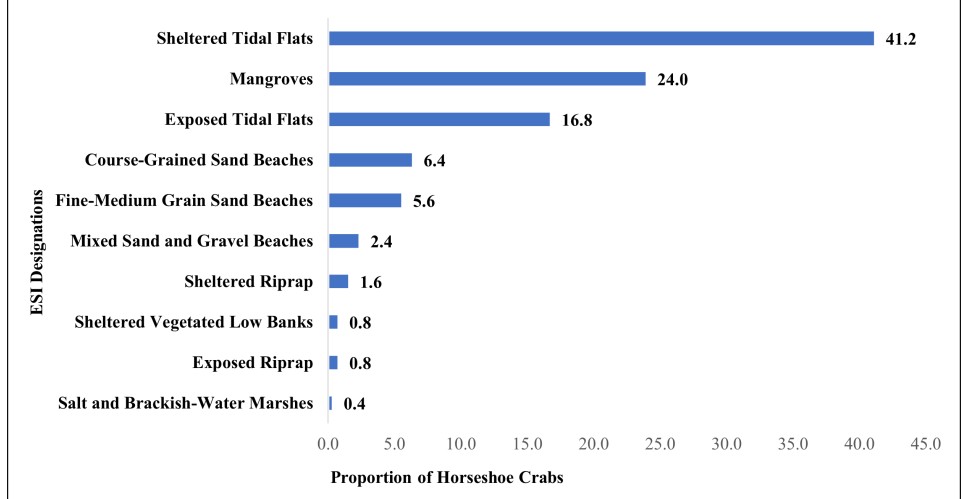

**Fig 3.** **Proportion of horseshoe crabs by ESI habitat types within the shoreline polygon of Pinellas County, Florida.**

The largest loss in Brevard County shoreline habitats associated with horseshoe crab spawning occurred from 2020 to 2100 (Table 3). Within the shoreline polygons, the habitat that lost the most area is fine- to medium-grain sand beaches (98%), followed by mangroves (94.8%), scrub-shrub wetlands (94.3%), riprap (84.4%), and sheltered vegetated low bank (71.2%).

## Discussion

This study determined that the habitats where horseshoe crabs were frequently observed in Florida's Pinellas and Brevard Counties were predicted to change under three sea-level rise scenarios. The continued rise in sea level directly threatens these spawning habitats and the future ecological and economic contributions of horseshoe crabs. This study characterized which habitat types and features were associated with spawning horseshoe crabs in Pinellas and Brevard County, Florida, and quantified the potential change of those habitats under conservative estimates of sea level rise from present day to 2100. These methods, combined with horseshoe crab spawning data, can be applied throughout the horseshoe crab range and can be used in consideration with shoreline armoring and restoration efforts.

Horseshoe crabs were predominantly sighted near sheltered tidal flats in Pinellas County and fine- to medium-grain sand beaches in Brevard County. These results are consistent with previous research that found juvenile horseshoe crabs occupy tidal flats during early life and that spawning horseshoe crabs favor sandy beach areas protected from wave energy, mainly between tidal flats and the extreme high tide water line [2,12]. The spawning beach habitats found in Pinellas and Brevard Counties were observed to have median slopes of 2° and 2.7°, respectively. The median slope of all habitat types combined was close to 0° because habitats such as mangroves, where a large proportion of horseshoe crabs were found in proximity, had a slope of 0°, which lowered the overall median. Previous studies have shown that horseshoe crabs prefer beaches with slopes ranging between 2°– 8° [12,16], where adequate aeration and moisture are favorable for embryonic development [12,15]. Since horseshoe crabs can become disoriented in flat areas after spawning, they will also rely on the slope of the beach rather than their vision to orient themselves and travel downslope [34].

Horseshoe crab sightings were most frequent around Fort De Soto Park, Honeymoon Island, and St. Petersburg in Pinellas County, in March and April. These spawning reports aligned with the timing of peak spawning in the nearby Florida counties of Pasco, Hernando, and Manatee [35]. Peak horseshoe crab sightings occurred in February and March in Brevard County, likely because of increasing water temperature, which triggers adults to move from deeper water to their

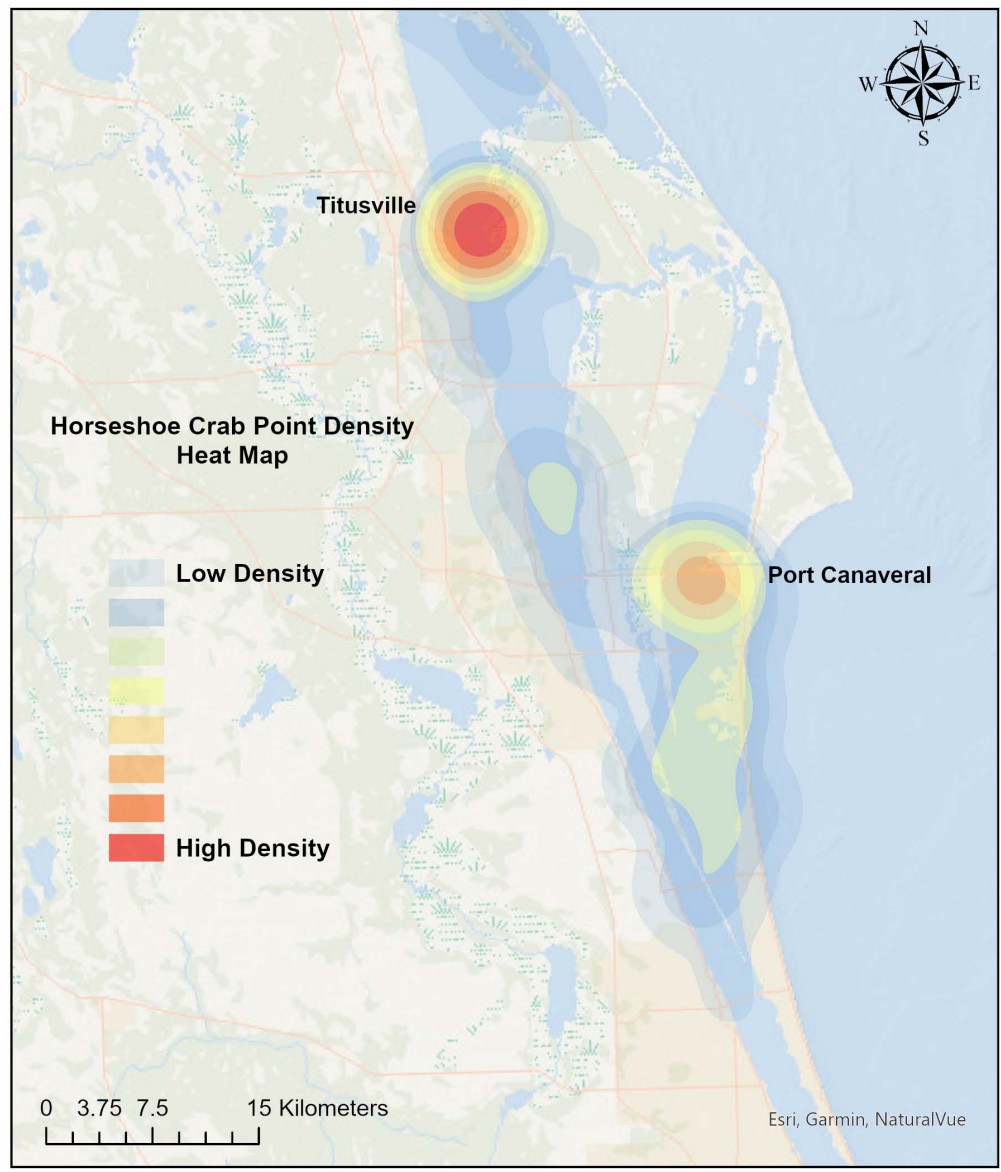

**Fig 4. Horseshoe crab spawning locations in Brevard County, Florida. Color denotes density of sightings in each location from 2020–2023.**

spawning sites [36]. In this study, the habitats associated with horseshoe crab spawning, such as tidal flats, mangroves, and fine- to medium-grain sand beaches, were those most affected by the sea-level rise scenarios.

While both counties had similar quantities of horseshoe crab sightings, the habitats that horseshoe crabs were sighted spawning were not consistent. This may indicate that horseshoe crab populations, which have been proven to be genetically distinct [37], have different habitat preferences [38]. Alternatively, shoreline development and armoring may vary between counties, and thus horseshoe crabs may be settling for suboptimal spawning habitats. In particular, Pinellas County has highly developed shorelines. This study determined that 23% of spawning horseshoe crab sightings in Pinellas County and 17% of spawning horseshoe crab sightings in Brevard County were closest to armored or hardened shorelines in their landward spawning designations. As sea levels rise, the sand patches near these structures that horseshoe crabs

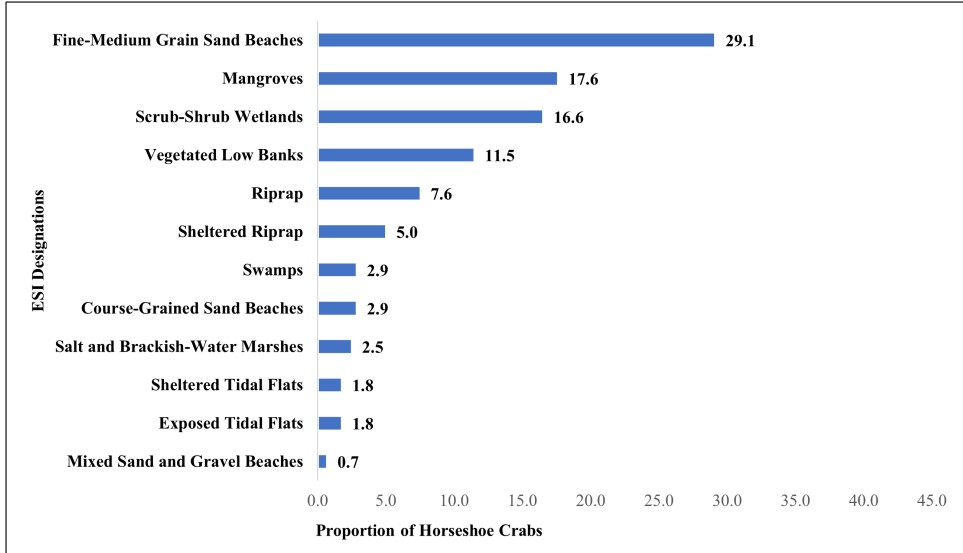

**Fig 5. Proportion of horseshoe crabs by ESI habitat types within the shoreline polygon of Brevard County, Florida.**

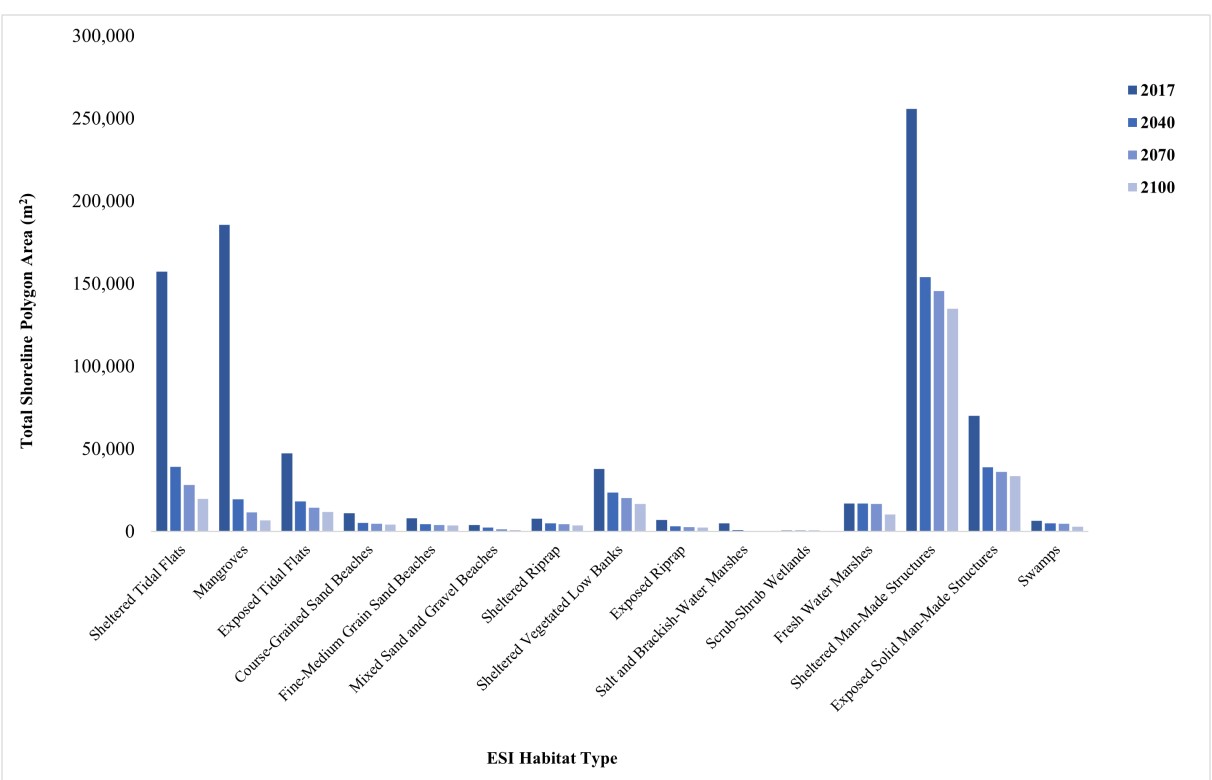

**Fig 6. Total area (m²) of each horseshoe crab habitat type projected to be present within the shoreline polygon in Pinellas County, FL in each sea-level rise scenario.**

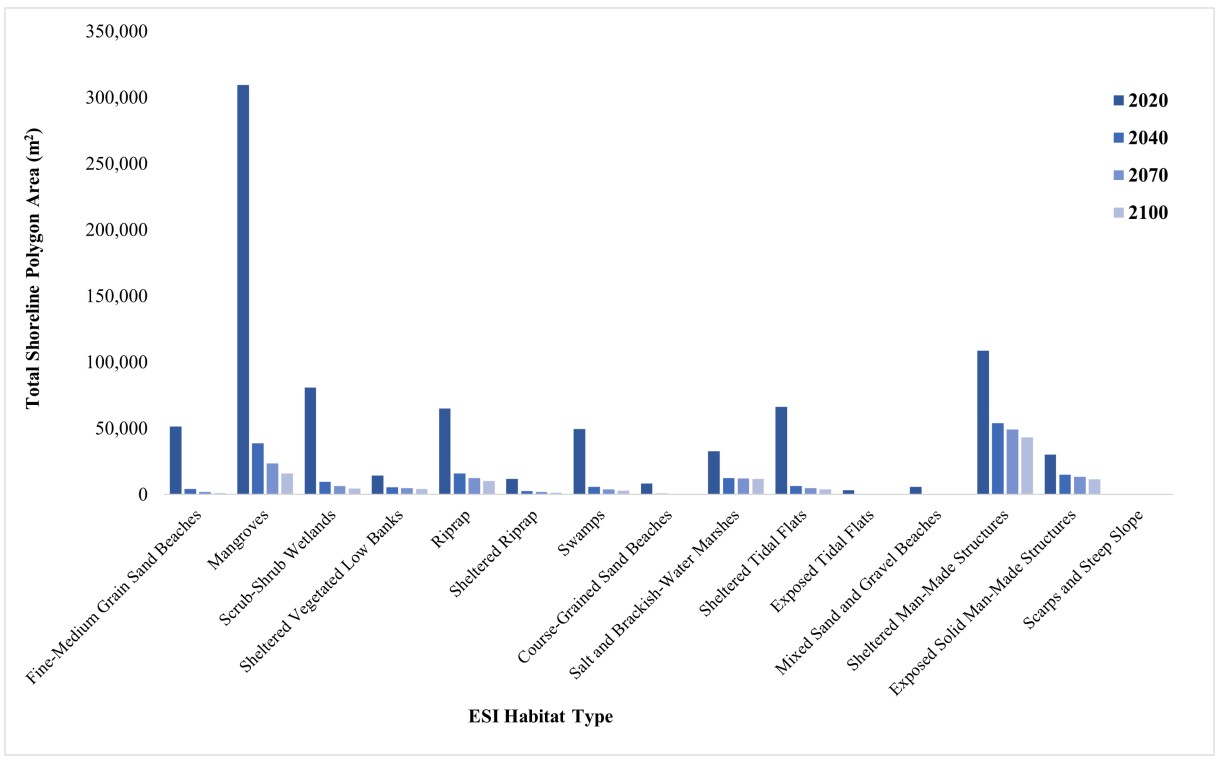

**Fig 7. Total area (m²) of each horseshoe crab habitat type projected to be present within the shoreline polygon in Brevard County, FL in each sea-level rise scenario.**

use for spawning will be inundated, and no longer available for spawning. Horseshoe crabs have adapted to changing environmental conditions over time and space, however, their ability to adapt may not be enough to overcome the challenges posed by armored structures such as bulkheads and seawalls [2]. These structures are designed to prevent shoreline erosion, but they can interfere with and prohibit the natural spawning habits of horseshoe crabs [16]. Living shorelines are often used as a natural alternative to hard-armoring techniques in areas where the shoreline is eroding [39]. This method uses natural materials such as plants, stones, oysters, and sand fills to protect, restore, or enhance the natural shoreline habitat [40]. By designing and implementing robust, long-term adaptation strategies, coastal communities and critical shoreline spawning habitats can be protected.

## Limitations

There are several limitations to the current study that should be considered in the interpretation of results. The nearest habitat to horseshoe crab sightings was based primarily on seaward designations. In cases where the seaward designation was absent, landward shore type was applied. As a result, in some instances the nearest habitat associated with horseshoe crabs was atypical and needed further review. If the ESI designation was not a habitat typically associated with horseshoe crab spawning, such as man-made structures or riprap, the site was reviewed in Google Earth. In some cases, small patches of sand adjacent to the structures were present, which may have been the actual spawning site but could not be identified by the ESI designations. Therefore, it is likely that man-made structures were classified as spawning habitats. Future studies may benefit from ground-truthing atypical spawning habitats. Since the sightings data is reliant on public reports, there may be a bias of sightings associated with more publicly accessible locations. Finally,

**Table 2. Potential loss of existing habitat based on the shoreline polygons associated with horseshoe crab spawning in Pinellas County from 2016–2100 due to sea level rise.**

| ESI Habitat Type | 2016–2100 Percent Area Loss Within Shoreline Polygons |
|---|---|
| Scrub-Shrub Wetlands | 24.2% |
| Fresh Water Marshes | 39.7% |
| Sheltered Man-Made Development | 47.2% |
| Sheltered Riprap | 50.7% |
| Exposed Solid Man-Made Development | 52.3% |
| Swamps | 54.3% |
| Sheltered Vegetated Low Banks | 55.7% |
| Fine-Medium Grain Sand Beaches | 56.6% |
| Course-Grained Sand Beaches | 63.2% |
| Exposed Riprap | 66.0% |
| Exposed Tidal Flats | 75.0% |
| Mixed Sand and Gravel Beaches | 79.9% |
| Sheltered Tidal Flats | 87.4% |
| Salt and Brackish Water Marshes | 94.9% |
| Mangroves | 96.3% |

**Table 3. Potential loss of existing habitat based on the shoreline polygons associated with horseshoe crab spawning in Brevard County from 2020–2100 due to sea level rise.**

| ESI Habitat Designation | 2020–2100 Percentage Area Loss Within Shoreline Polygons |
|---|---|
| Sheltered Man-Made Development | 60.3% |
| Exposed Solid Man-Made Development | 61.7% |
| Salt and Brackish Water Marshes | 63.4% |
| Sheltered Vegetated Low Banks | 71.2% |
| Riprap | 84.4% |
| Sheltered Riprap | 87.9% |
| Exposed Tidal Flats | 92.7% |
| Swamps | 94.1% |
| Sheltered Tidal Flats | 94.2% |
| Scarps and Steep Slope | 94.3% |
| Scrub-Shrub Wetlands | 94.3% |
| Mangroves | 94.8% |
| Course-Grained Sand Beaches | 95.5% |
| Mixed Sand and Gravel Beaches | 97.8% |
| Fine–Medium Grain Sand Beaches | 98.0% |

this study did not analyze potential habitat gain or examine potential habitat changes. These percentages may not indicate absolute loss because habitat changes are likely to occur. However, it demonstrated the potential loss of a specific spawning habitat that was used by horseshoe crabs. Identifying the spawning habitats that may not suffer complete loss to sea level rise can narrow the focus in predicting whether they will remain the same or transition to another habitat type. It is also important to consider the habitat changes that may occur on the periphery of frequently used spawning habitats, as this could either hinder or enhance spawning success. For example, if a spawning site becomes unfavorable, an alternative habitat may become more suitable, enabling horseshoe crabs to spawn in that area instead [38].

## Habitat changes

The natural movement of beaches over time can be restricted by hard structures adjacent to the coastline [41]. Developed coastal ecosystems are less capable of dynamically responding to sea level rise compared to natural areas, where beaches could adjust by changing landward to maintain equilibrium [42]. Several studies have begun investigating the consequences of rising sea levels on different habitats and their likelihood of changing to other habitat types. One such study conducted by Geselbracht et al. [43] focused on six estuaries along the Gulf Coast of Florida. The study employed a Sea Level Affecting Marshes Model (SLAMM) to predict the possible changes in habitat until 2100 and considered three scenarios with varying degrees of sea level rise (0.7 m, 1 m, and 2 m). According to their findings, if sea levels rise by 1-meter, tidal flats will likely lose the most spatial extent of all coastal habitats in the model. In contrast, mangrove forests are expected to increase their spatial extent in the first two sea-level rise scenarios (0.7 m and 1 m). However, if sea levels rise by 2 meters, mangroves will lose their total area to open water. This demonstrates a threshold beyond which they will be unable to cope with the rising sea levels. Therefore, further research is needed to determine accretion rates for various habitat types and changes that may occur in specific areas where horseshoe crabs spawn. If changes occur, identifying habitats that promote or deter successful egg development will aid in protecting shorelines and preserving horseshoe crab populations. Several studies have demonstrated that horseshoe crabs are not limited to sandy beaches and have been observed to spawn in alternative habitats such as peat beds, muddy sediments, and fringing salt marshes [38]. Although these habitats were previously thought to be unsuitable for embryonic development [16], recent studies have shown that clutches of horseshoe crab eggs laid in muddy salt marsh habitats contain similar developmental stages as clutches laid on sandy beaches [11,38].

## Conclusion

Horseshoe crabs have adapted to changing environmental conditions over millions of years [44]. However, it is important to consider the different rates of change at which populations will be affected before assuming they will continue to persist [44]. The potential consequences of rising sea levels can be far reaching and can affect many species, especially those with life history strategies such as horseshoe crabs. This paper proposes a practical approach to using available data on species presence and characterizing shoreline habitats that are critical and likely to be impacted by sea level rise. Managers can use these projections to identify spawning habitats that are likely to change over the next 100 years. This information can help in making well-informed decisions regarding the best locations of living shorelines and beach replenishment programs. Such initiatives can support and mitigate the effect of rising sea levels, safeguard coastal communities, secure the survival of keystone species, and promote a healthy and sustainable ecosystem.

## Supporting information

**S1 Fig. Screenshot of horseshoe crab sighting questionnaire.**
(PDF)

**S1 Data. HSC data from Survey123 website (2009 - 2023) Edited.**
(XLSX)

## Acknowledgments

This manuscript was made possible by the support and collaboration of H. Jane Brockmann, Casey Butler, Ryan Gandy, Katharine Becker, and Kara Radabaugh. We are grateful for the exchange of ideas within our community. Thank you to the Citizen Scientists of Florida Horseshoe Crab Watch and the public for reporting spawning horseshoe crabs.

## Author contributions

**Conceptualization:** Danielle Contrada, Claire Crowley-McIntyre, Berlynna Heres.

**Data curation:** Danielle Contrada, Berlynna Heres.

**Formal analysis:** Danielle Contrada.

**Funding acquisition:** Claire Crowley-McIntyre.

**Investigation:** Danielle Contrada.

**Methodology:** Danielle Contrada.

**Project administration:** Danielle Contrada, Claire Crowley-McIntyre.

**Resources:** Danielle Contrada, Claire Crowley-McIntyre, Berlynna Heres.

**Supervision:** Claire Crowley-McIntyre.

**Validation:** Danielle Contrada.

**Visualization:** Danielle Contrada.

**Writing – original draft:** Danielle Contrada.

**Writing – review & editing:** Danielle Contrada, Claire Crowley-McIntyre, Berlynna Heres.

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
