## [Decision Letter · Decision Letter 0]

13 Dec 2024

Dear Dr. Heres,

Thank you for submitting your manuscript to PLOS ONE. After careful consideration, we feel that it has merit but does not fully meet PLOS ONE’s publication criteria as it currently stands. Therefore, we invite you to submit a revised version of the manuscript that addresses the points raised during the review process.

Indicate which changes you require for acceptance versus which changes you recommendAddress any conflicts between the reviews so that it's clear which advice the authors should followProvide specific feedback from your evaluation of the manuscript

We look forward to receiving your revised manuscript.

Kind regards,

Phuping Sucharitakul

Academic Editor

PLOS ONE

4. We note that Figures 1, 2 and 4 in your submission contain [map/satellite] images which may be copyrighted. All PLOS content is published under the Creative Commons Attribution License (CC BY 4.0), which means that the manuscript, images, and Supporting Information files will be freely available online, and any third party is permitted to access, download, copy, distribute, and use these materials in any way, even commercially, with proper attribution. For these reasons, we cannot publish previously copyrighted maps or satellite images created using proprietary data, such as Google software (Google Maps, Street View, and Earth). For more information, see our copyright guidelines: http://journals.plos.org/plosone/s/licenses-and-copyright.

1. You may seek permission from the original copyright holder of Figures 1, 2 and 4 to publish the content specifically under the CC BY 4.0 license. 

5. We note that there is identifying data in the Supporting Information file < HSC data from Survey123 website (2009 - 2023) copy.xlsx>. Due to the inclusion of these potentially identifying data, we have removed this file from your file inventory. Prior to sharing human research participant data, authors should consult with an ethics committee to ensure data are shared in accordance with participant consent and all applicable local laws.

-Location data

Reviewers' comments:

Reviewer's Responses to Questions

**Comments to the Author**

1. Is the manuscript technically sound, and do the data support the conclusions?

Reviewer #1: No

Reviewer #2: Partly

2. Has the statistical analysis been performed appropriately and rigorously?

Reviewer #1: I Don't Know

Reviewer #2: No

3. Have the authors made all data underlying the findings in their manuscript fully available?

Reviewer #1: Yes

Reviewer #2: Yes

4. Is the manuscript presented in an intelligible fashion and written in standard English?

Reviewer #1: No

Reviewer #2: No

Reviewer #1: This manuscript by Contrada et al. seeks to assess “the potential impact of sea-level rise on horseshoe crab nesting habitats”. The authors do this by combining sightings of horseshoe crab spawning with GIS characterization of how shorelines differ across sea level rise scenarios, based on current habitat characterizations. The findings may be relevant if presented in the proper contexts and if the authors restrict their discussion to the results of their analyses. The methods of the paper are also unclear, but I think they can be improved with greater organization and a clearer workflow of the process. My specific comments below are meant to be helpful in providing constructive feedback to improve this manuscript.

• L27: There are no data on living shorelines presented in this paper to support this conclusion. Please remove.

• L40: Add “hereafter ‘horseshoe crab’” so the reader knows that you will be using that name to refer to this species.

• L51: Which changes are the authors referring to when they mention “these changes”? Please be specific about which changes you are referencing.

• L90: Only the rufa subspecies relies on horseshoe crab eggs for their migration.

• L103: Please clarify what the symbol “±” is meant to represent. Presumably this is a measure of error, but is it the standard error of the mean, or some other measure?

• L105: Please provide a link or another citation so readers know how to locate this information.

• L120-123: The font in this section is different than surrounding text.

• L142-143: This sentence indicates that the present study used only data from 2017-2023, but the sentence before includes reference to a figure with data from 2002-2023. Please clarify the use of these two datasets in the present study (i.e., how and why the data from 2002-2023 are used, and how and why the data from 2017-2023 are used.

• L143-144: Please provide specifics on the type and timing of imagery used to develop to ESI layers used in your analysis. This is important context for understanding when your habitat characterizations took place as compared to when your horseshoe crab sighting information was collected.

• 143-155: Maybe create a separate section for habitat designation (e.g., ‘Habitat Characterization’, or something similar) where you can describe the ESI? Right now this information is included in the section of ‘horseshoe crab nesting data’.

• L170-180: This paragraph would also fit better in a ‘Habitat Characterization’ section.

• L191-194: Please include specific information on which analyses were used.

• L189-196: This section currently comes after the sentence stating that “habitats ranked “zero”, which classified areas where nesting was possible, were used for further analysis”. So were the horseshoe crab reports discussed here (L189-196) all of the horseshoe crab reports, or just the ones ranked as zero? I think this is referring to all the horseshoe crab reports, but the way this is currently written I am unsure. Please clarify your workflow.

• L230: Kernal density analysis is not discussed in your methods. Please include information in your methods and workflow regarding what this analysis is and how it was conducted.

• L248: Kernal density is again mentioned here but not in the Methods, please address.

• L238-241, L255-259, L280-282: Why are figure captions located in the middle of the text?

• L308: Please clarify why is meant by ‘economic success of horseshoe crabs’.

• L310: Changing ‘shift’ to ‘loss’ would be more accurate. The results quantified the potential loss of habitats. A ‘shift’ in habitat would suggest that you quantified how habitat changed from one habitat to another.

• L348-360: The results provide no information on horseshoe crab eggs or their use by predators such as shorebirds. This section of the discussion is not relevant, but L337-339: Redundant with introduction.

• L373: The authors here acknowledge that they did not “examine potential habitat shifts”, an important point that I am glad they acknowledge here. I implore the authors to review their own language throughout the rest of the manuscript to be clear that they are not suggesting this elsewhere in the manuscript.

• L405-427: Living shorelines are not in the results of the manuscript. They are not tested experimentally nor do they appear to be listed as one of the habitat types in the analyses. The discussion of living shorelines thus serves purely a review of published studies, none of which appear to be specifically related to horseshoe crab habitat. As such, this section is not relevant to the results of the study and should be removed from the manuscript.

• L429: You use the term ‘horseshoe crab’ throughout the manuscript to refer to L. polyphemus, but in this instance you are using ‘horseshoe crab’ to refer to a broader taxonomic group of horseshoe crabs over millions of years. This should be clarified. Maybe you could change this to say ‘Xiphosurid horseshoe crabs have adapted to changing environmental conditions over millions of years’?

• L434-435: This statement is inconsistent with the limitations described by the authors. Please re-frame this statement to acknowledge the uncertainties around how these findings might be used by managers.

• L441: Is there acknowledgement of the funds that supported personnel to conduct this work or to collect horseshoe crab data?

• L452: Publication is missing

• L518: Is this a publication or a website? More information is needed.

• L615-617: This is not peer-reviewed. Please cite peer-reviewed study. Maybe this one?

https://esajournals.onlinelibrary.wiley.com/doi/full/10.1002/fee.2738

• Figures are blurry and hard to read.

• Figure 5: Please add a x-axis with values

• Figures 6 & 7. These are difficult to view for color-blind readers

Reviewer #2: Synopsis

In this manuscript, the authors characterized the possible change in nesting habitat at where American horseshoe crab were observed nesting in FL. Using crowd-sourced, community science data, the authors generated a dataset of HSC observations, assigned landcover classifications where HSC were observed, and quantified loss of landcover classifications at three future timesteps. The authors found habitat where HSC were observed in 2017 and 2020 was predicted to be lost due to anticipated sea level rise. The authors propose that living shoreLmay be a more suitable climate adaptation than hardened structures, such as sea walls, as living shoreLare also beneficial for spawning HSC. I see value to this work and highly encourage the authors to pursue revision of this work.

In the manuscript’s current form, there is inconsistent and unclear use of terminology that makes methodology unclear, and additional clarification is necessary to fully comprehend the study findings. Below, I’ve attempted to provide thorough feedback in the spirit of facilitating revision.

General feedback

-Consider writing in first person (e.g., we studied, we collected data, we analyzed, we found) throughout.

-Consistency in terminology: Throughout the manuscript, there’s inconsistent use of terminology that makes it difficult to follow the authors’ message. Some terms, like landward and seaward, should be clarified on their first use. There’s inconsistent formatting of name of horseshoe crab-consider dropping American after first use. Please consider using anthropogenic instead of ‘man-made’ throughout. There’s discussion on habitat shift, though what a ‘shift’ is was not defined, and there’s contradictory statements in the manuscript as to whether evaluating shifts occurred (see L337-374 vs L116-118, 126-129, 170-172, 308-311; I do not believe ‘shifts’ were measured, as this, to me, suggests an area of landcover classification x transforming to landcover classification y, though the methods are unclear). Please consider using change instead of loss when quantifying the area of landcover classifications predicted to be affected by sea level rise.

-Introductory content: I found it unusual to read a horseshoe crab paper that does not use the word ‘spawning’ to describe horseshoe crab breeding behavior. I recommend using the word ‘spawning,’ when referring to active breeding behaviors, and ‘nests,’ to refer to the locations where HSC eggs are deposited.

-Methods and habitat selection/preference: This manuscript presents a characterization of the areas where horseshoe crabs were reported. There is no use vs availability analysis, which is an analysis frequently used to explore organismal habitat selection (and preference). Without a use vs availability analysis or a similar statistical analysis, the use of ‘habitat selection’ or ‘preference’ is inappropriate.

-Community science: There’s a lot of excellent value from crowd-sourcing data collection, though there can be some caveats to consider. As written, I’m unclear on the potential caveats and more information is needed to understand the data that was submitted by community members. I’ve outlined some points for consideration in the manuscript.

--Consider adding a paragraph later in the Introduction that explains the HSC community science program in FL, then get into the details of this program in the Methods. It would be helpful to describe the value of using community science data in evaluating species distributions. The end of the Introduction would likewise be an excellent place to mention the value of the community-derived HSC database.

--What is meant by ‘horseshoe crab activity’ (L134-135)? Does this mean that when community members saw a spawning horseshoe crab (as suggested by header on line 133), they reported, or was the data portal intended for any observation of a crab? If observations were of any crab, how was nesting distinguished from a dying or deceased crab on the beach, or a crab that may be embedded in the substrate between tides?

--Were observations only of marked crabs (e.g., https://www.fws.gov/project/horseshoe-crab-cooperative-tagging-program) or did observations also include unmarked crabs? If the dataset included unmarked crabs, how did the authors control for the potential of repeat observations?

--Were there efforts to control for sampling effort (frequency of reporting) which could vary based on popularity of the beach with recreationists/community members, or in areas that may be more popular with nature-minded individuals that would be more inclined to use a wildlife reporting app?

--What training resources were available for the public? For example, the data include fields for HSC sex, which may not be readily apparent to community members. Please consider explaining signage, visuals of the app as an appendix, and other resources that community members could use when submitting a report.

-Use of citations: Some sentences in the manuscript appear to be missing citations (e.g., L69-70). Numerous review articles are cited in lieu of the primary sources; please consider citing primary articles. Some citations appear to be used incorrectly (e.g., reference [21] in L92-94). Please consider a careful review of references to ensure all are used appropriately.

Specific feedback

-L1-2: The use of populations in the title suggests an evaluation of demographic impacts, which were not part of the study objectives. Not evaluating demography is not a detractor, though perhaps a recast to reflect that this manuscript evaluates projected habitat changes for spawning HSC would be more appropriate. Consider including full study organism name (American horseshoe crab) and Latin binomial epithet.

-L12: Sentence starts with a value-laden word. Consider replacing ‘unfortunately’ with ‘Though.’

-L13 and throughout: Lowercase H and C in horseshoe crab. After first mentions of American horseshoe crab in Abstract and Introduction, consider consistently using ‘horseshoe crab.’

-L17 and throughout: Consider replacing ‘habitat type’ from ESI products with ‘landcover classification,’ re: Krausman, P. R. and M. L. Morrison. 2016. Another plea for standard terminology. The Journal of Wildlife Management 80(7):1143-1144. doi: 10.1002/jwmg.21121

-L32-41: This paragraph about climate change indicates that sea level rise will affect coastal ecosystems, but does not provide detail on what changes sea level rise is predicted to cause. Describing anticipated consequences from sea level rise will be important for understanding the potential implications for HSC.

-L36-38: Consider removing ‘in response to the threat of rising sea levels,’ as the starting clause of the sentence already indicates this.

-L42 vs 148: rip rap vs riprap.

-L49-50: HSC also breed along the Yucatan Peninsula of Mexico, though the Mexican population is genetically distinct (García-Enríquez, J. M. et al. 2023. Genetic study of the American horseshoe crab throughout its Mexican distribution. Conservation and management implications. Biodiversity and Conservation 32:489-507. doi: 10.1007/s10531-022-02508-4).

-L58-59: The background information on HSC breeding ecology is scattered in the Introduction. It would be helpful to describe HSC life history, spawning behavior, and development in a single, cohesive paragraph, and this location would be an appropriate location.

-L59-61: Consider a recast for this sentence. Many organisms will breed/reproduce in suboptimal areas if preferred habitat is unavailable (Fretwell, S. D. 1972. Populations in a seasonal environment. Princeton University Press.) and there’s a lot of work in understanding population density, age, and other factors in habitat preference and habitat selection. Perhaps recast to focus on habitat features that facilitate HSC spawning and embryonic development. Consider moving a paragraph from the Discussion (habitat shifts paragraph on L382-404) into the Intro, which feels more appropriate when considering beneficial habitat features for HSC.

-L65-67: Consider adding “respectively” after 0.3 mm [12] to clarify measures correspond to each state.

-L90: Consider Red Knot Calidris canutus rufa, the subspecies that migrates from South America to Arctic Canada and relies on HSC eggs in DE bay.

-L92-94: There’s limited evidence to suggest HSC are a major dietary item of sea turtles since the 1980s. The reference used here [21] does not support HSC are a dietary item of sea turtles.

-L96-112: While indicating HSC importance is valuable, this paragraph contains a lot of information on the economic value of HSC, which feels tangential to the study goals. Consider shortening this section to succinctly highlight the economic value of HSC, and incorporate threats to HSC.

-L99: Consider Latin binomial epithets for American eel and whelk.

-L103: Please clarify measure after average-is this SD, SE?

-L105: Citation should be numeric.

-L109: Remove space between ‘sea-’ and ‘level’

-L115: Study does not truly evaluate habitats, as this suggests a statistical comparison. Instead, the manuscript presents a characterization of the habitats and simulates a projected change at three distinct times. Please consider recasting to clarify this is a descriptive study.

-L116-118: Why were these years chosen? Please consider including a rationale for these years.

-L119: Introduction seems to end abruptly. Please consider including statements on the importance of this work for FL, and how these results could extend elsewhere along the Gulf and Atlantic coasts.

-Methods line 120: Consider placing study site description before statement on ArcGIS-i.e., reorganize Methods with L125-132, L133-168, then include L121-123 between L169-170.

-L121: Add manufacturer information (e.g., Esri, Redlands, CA, USA).

-L126: Remove Latin epithet for horseshoe crabs.

-L130: Replace text citation with numeric.

-L126-132: Please consider adding additional detail on study sites, as this will be useful for others in both contextualizing findings and understanding how results here may extend elsewhere. Consider adding approximate coordinates for locations (can be centered in county) for readers. Consider describing the usage of shorelines-how much of the area is protected (e.g., wildlife refuges), what’s the ratio of protected to unprotected areas (e.g., space center in Brevard county). Additionally, this study includes areas on the Gulf and Atlantic coasts, which may have very different temperatures, currents, and habitats. The authors may consider framing this study as an Atlantic coast/Gulf coast comparison, which may expand the interest of readers outside FL.

--Relatedly in this section, please consider a separate map that shows study areas, and limit locations on the map to only include names. The information displayed in Figure 1 is a result and would be more suitable in the Results section.

--Consider explaining how sea level rise is expected to affect the barrier islands in the study areas. Are barrier islands migrating, changing?

-L146-147: A brief explanation of how coastal habitat designations were made in ESI would be useful. E.g., were these designations the result of supervised classifications in ArcGIS, were they

-L134-135, Figure 1: Shows a result, would be more appropriate to reference in the Results section and not in Methods.

-L134-135: Who were results reported to (i.e. which agency)? How were people able to complete the form (e.g., was there signage, or did people have to know of the reporting system) and how did they know how to fill out fields in the form? Understanding the resources and training available to respondents is necessary for understanding the validity of behavioral observations that include fields for HSC sex. Are these of marked HSC (e.g., https://www.fws.gov/project/horseshoe-crab-cooperative-tagging-program)? Are reports of HSC standardized to control for ‘sampling effort,’-e.g., I would a priori predict more reports of HSC in areas with more people. If HSC are unmarked, how might the possibility of repeated observations of the same individual in areas with high human traffic skew interpretation of results?

-L138: How was it determined if an observation was ‘robust’?

-L141: Include a reference for kernel density analysis.

-L142: How was subtidal nesting ruled out from the data? From a quick look at the data, I did not see a field explicitly for spawning/nesting. How was it determined if an observation was below the water line? This is especially important, as tideLcan change rapidly within a year, and vary drastically between years.

-L143-149: Additional information describing the ESI dataset is necessary. How did NOAA designate landcover classifications? Please consider including a table that lists the landcover classifications (habitat types) and a description of that classification.

-L160-161: Were in-person or spatial assessments of HSC observations at high tide or low tide? Were assessments made at the same year as the reported observation? How did the authors account for potential changing shorelines?

-L165-166: It’s unclear what kinds of landcover classifications were considered, and so it remains unclear what habitats were considered likely, unlikely, or unknown nesting substrates. Please include a table of landcover classifications, a description of that classification, and sources that support assignments in the ranking system.

-L178-180: It was surprising to see HSC do not nest on FL’s Atlantic and Gulf coast facing beaches; elsewhere on the North Atlantic, HSC nesting can occur on Atlantic beaches. A citation to help justify the exclusion of these beaches in this study would be useful.

-L182, 206: There are two subheaders for ‘Sea-Level Rise Scenarios,’ in the Methods section.

-L186-188: The Atlantic Coast and Gulf Coast are anticipated to experience differing extents of sea level rise. Were the study areas evaluated under the same scenario, or were scenarios spatially explicit?

-L190-193: The explanation behind the choice to use 0.6 m wide polygons suggests a possible misinterpretation of sea level rise. If I understand, text appears to suggest that a 0.6 m strip is a conservative boundary that encompasses shoreline habitats that will be affected by a 0.5 m sea level rise. However, this would imply that a 0.5 m sea level rise will result in a waterline shift of 0.5 horizontal m across the beach, when in practicality, the waterline should shift 0.5 m in elevation. To understand the land area affected by the elevational change of the waterline, analyzing the DEM is necessary to determine that line.

-L199-204: I’m unclear what additional information is gained from the inclusion of nesting habitat slope. Was there one DEM available to quantify slope, and if so, how did the authors consider the potential for annual variation in beach slope and can slope be reasonably extrapolated to 2040, 2070, and 2100 to inform estimates of habitat change due to sea level rise? Were slope estimates averaged across all habitat types in an area? It’s also impossible to glean HSC selection of slope in this study without a use vs availability analysis.

-L207-209: Consider moving reference to the digital elevation model to line 199, which is the first reference of the DEM in the Methods. Greater description of the DEM would be useful.

-L210-212: It’s unclear if the timesteps reflect a 2D area of change (‘subtracting’ the area anticipated to be lost to sea level rise from the coast at time step t) vs a 3D area of change that considers the volume of water and the beach slope. Additional clarification is necessary to understand how pixels were considered ‘underwater’ at each of the three timesteps.

-L214: Remove extra apostrophe between quotes of “Erase”

-L217-218: Consider omitting how percent is calculated.

-~L222: Consider providing some general summary statistics about the study areas-lengths of shorelines, annual areas of landcover classifications. Consider also including summary statistics on community science reporting, which could be used to explain Figure 1 (vs in the Methods). On average, how many observations are made per year (± SD)? In Fig 1, how many years of data are shown? Would results change if data were presented as mean annual crab sightings? The Introduction notes two distinct spawning periods for FL HSC (Feb-May, Aug-Nov; L84-85), which is very unique. Please consider describing temporal variability in HSC nests observations.

-L225, 243: Separating results into counties makes this paper very FL-specific, and difficult to glean generalities that extend beyond FL. Consider recasting to focus on Gulf coast vs Atlantic coast, and characterize the habitats available on each coast.

-L233-236, 250-253: How were these habitats (landcover classifications) delineated? This wasn’t clear in the Methods

-Fig 2, 4: I note that some of these ‘clusters’ overlap with densely populated or popular areas (e.g., St Petersburg). How were data handled to account for sampling effort? Consider combining this figure into one with two panels.

-Fig 3, 5: These show percents, not proportions. Consider combining this figure into one with two panels.

-Table 2, 3: Consider including a starting area of the landcover classification from 2017/2020. As an example, the projected loss of 94.8% of mangroves appears alarming, though may be of minimal ecological consequence if the starting area of mangroves represented an insignificant area in 2020.

-Results: Consider including tables for each county (consider presenting as Atl and Gulf coasts) with numbers of reported crabs in each landcover classification, and/or numbers of crabs per year.

-L304-313: This line buries the lede by restating results. Consider recasting to focus on the most important finding, that habitats where HSC were observed most frequently were predicted to be lost under sea level rise scenarios. Then contextualize this finding.

-L328: Capital S in De Soto Park

-L337-374: This sentence is at odds with statements in the objectives (L116-118), methods (L1226-129, 170-172), and the discussion (L308-311).

-L336-337: The use of habitat preference is not appropriate here, as there was no use vs availability analysis. Further, there was no contextualization of Atlantic coast vs Gulf coast, or ocean vs bay areas, which prevents inference on preference.

-L336-347: How does the area of hardened and developed shoreLin the two study counties compare to other areas along FL’s coast?

-L362-363: Consider removing this sentence.

-L348-360: This section feels out of place, especially considering that temporal variation of results was not presented. Consider removing.

-L361-381: Consider recasting the Discussion, and incorporating this information, as relevant, in Methods and Discussion. While it is critical to transparently acknowledge limitations of any given study, singular dedicated paragraphs can be unintentionally disparaging. Please consider contextualizing results and within that same paragraph, provide the relevant limitations for that finding.

-L443-434: The objectives of this study did not include providing a practical approach for identifying areas with high concentrations of species that rely on shoreline habitats for nesting, but rather investigated the potential impact of sea level rise on HSC nesting habitas (L113-114). The methods used did not appear to prioritize elucidating high concentrations of HSC, as the authors note they chose to focus on two counties (L116). Further, the methods are unclear if the analysis accounted for sampling effort, which would be needed to fully understand species abundances and distributions.

-Data: The data submitted with the manuscript should include metadata that define each of the columns. But concerningly, the data contain sensitive and personally-identifying information. I recommend removing columns U through AI, which contain sensitive information.

**Do you want your identity to be public for this peer review?** For information about this choice, including consent withdrawal, please see our Privacy Policy

Reviewer #1: No

Reviewer #2: No

---

## [Author Response · Author response to Decision Letter 1]

16 Jul 2025

We have attached our response to reviewers. We found the critique extremely helpful and we appreciate their efforts.

---

## [Decision Letter · Decision Letter 1]

10 Aug 2025

Dear Dr. Heres,

I think your manuscript is nearly ready for publication. Although one of the reviewers recommended a major revision, I believe only minor revisions are needed. If you make the required changes, I will be happy to accept it for publication.

We look forward to receiving your revised manuscript.

Kind regards,

Phuping Sucharitakul

Academic Editor

PLOS ONE

Journal Requirements:

Reviewers' comments:

Reviewer's Responses to Questions

**Comments to the Author**

Reviewer #1: (No Response)

Reviewer #2: All comments have been addressed

2. Is the manuscript technically sound, and do the data support the conclusions?

Reviewer #1: No

Reviewer #2: Yes

3. Has the statistical analysis been performed appropriately and rigorously?

Reviewer #1: I Don't Know

Reviewer #2: Yes

4. Have the authors made all data underlying the findings in their manuscript fully available?

Reviewer #1: Yes

Reviewer #2: Yes

5. Is the manuscript presented in an intelligible fashion and written in standard English?

Reviewer #1: No

Reviewer #2: Yes

Reviewer #1: The revised manuscript by Contrada et al. describes the potential impacts of sea level rise on coastal habitats used by spawning horseshoe crabs. The authors combine sightings of horseshoe crab spawning with sea level rise scenarios and shoreline characterization. Unfortunately, the authors chose not to incorporate a number of edits suggested by the previous reviews, which continue to make the manuscript difficult to interpret. Most importantly, this is related to inconsistent terminologies. A few specific items are mentioned below.

• L137-141: If I’m reading this correctly, Figure 1 represents information that was compiled and presented for this article and has not previously been published. As such, it would be more appropriate for Figure 1 to be considered a result, and should be presented in the Results section. In this way, the author can then clarify which datasets were used for which portions of this current study.

• Please include justification for how calculating slope advances the stated goal of the study describing the “potential impact of sea-level rise on horseshoe crab spawning”?

• L377: Suggest using ‘anthropogenic’ instead of man-made.

• L380-381: The authors are quoted as saying “Finally, this study did not analyze potential habitat gain or examine potential habitat shifts”. Both reviewers from the previous version of the submitted manuscript pointed out the inconsistency in terminology and requested more consistency. Given the authors own words here, ‘habitat change’ would be an appropriate phrase to use instead of ‘habitat shift’.

• L387-388: The cited study doesn’t seem to support the statement. If I’m reading it correctly, the paper doesn’t address a shift in spawning areas.

• L360-361: I believe this statement overstates the benefits of living shorelines. Please provide a citation for this statement: “One effective solution to protect shorelines from the impact of rising sea levels is to implement “living shorelines”.

Reviewer #2: PONE-D-24-52275R1

Potential implications of rising sea level on American Horseshoe Crab (Limulus polyphemus) spawning beaches in two Florida counties

In this manuscript, the authors characterized the possible change in spawning habitat at where American horseshoe crab in FL. Using crowd-sourced, community science data, the authors generated a dataset of HSC observations, assigned landcover classifications where HSC were observed, and quantified loss of landcover classifications at three future timesteps. The authors found areas of spawning habitat where HSC were observed in 2017 and 2020 was predicted to be lost due to anticipated sea level rise. The authors propose ways their work can inform restoration and management efforts, and how methodology could be used elsewhere.

I appreciate the authors' care and dedication during revision, which has improved their interesting and valuable work. I have very minor areas of feedback:

-Contextualization of sites: I understand the authors' decision to retain reference to FL counties (vs suggestion to reframe as Atlantic coast/Gulf coast). In Methods (~lines 129-132), please consider adding a brief clause indicating that HSCs in these sites are not expected to differ significantly in behavior (as noted in the response to reviewers document, and cite any supporting information that already exists in the Lit Cited section). Doing so will help reinforce that these are analagous sampling points and not necessarily points chosen to represent different populations/climate change regimes.

-Line 129: Unnecessary 'n' in sentence.

-Lines 118, 133, 225, 232, 348, 356: Unusual punctuation (e.g., space before period, two periods, missing period, missing space, missing period). Please consider a careful review to check for other typos.

-Lines 442-444: I believe the complete citation for this should be: Sweet WV, Kopp RE, Weaver CP, Obeysekera J, Horton RM, Thieler ER, Zervas C. 2017. Global and regional sea level rise scenarios for the United States (Technical Report NOS CO-OPS 083). National Oceanic and Atmospheric Administration, Silver Spring, MD. doi: 10.7289/V5/TR-NOS-COOPS-083

**Do you want your identity to be public for this peer review?** For information about this choice, including consent withdrawal, please see our Privacy Policy

Reviewer #1: No

Reviewer #2: **Yes: ** Christy N Wails

---

## [Author Response · Author response to Decision Letter 2]

12 Sep 2025

Our response to reviewers can be found in the response to reviewer document attached. We found the advice very helpful and are grateful for their time and effort.

---

## [Editor Report · Decision Letter 2]

18 Sep 2025

Potential implications of rising sea level on American Horseshoe Crab (Limulus polyphemus) spawning beaches in  two Florida counties

PONE-D-24-52275R2

Dear Dr. Heres,

We’re pleased to inform you that your manuscript has been judged scientifically suitable for publication and will be formally accepted for publication once it meets all outstanding technical requirements.

Kind regards,

Phuping Sucharitakul

Academic Editor

PLOS ONE
---

## [Editor Report · Acceptance letter]

PONE-D-24-52275R2

PLOS ONE

Dear Dr. Heres,

I'm pleased to inform you that your manuscript has been deemed suitable for publication in PLOS ONE. Congratulations! Your manuscript is now being handed over to our production team.

Kind regards,

on behalf of

Dr. Phuping Sucharitakul

Academic Editor

PLOS ONE